# Use of In Situ Soil Solution Electric Conductivity to Evaluate Mineral N in Commercial Orchards: Preliminary Results

**Elena Baldi** [1,*] , **Maurizio Quartieri** [1] **, Enrico Muzzi** [1] **, Massimo Noferini** [2] **and Moreno Toselli** [1]

[1] Department of Agricultural and Food Sciences, University of Bologna, Viale Fanin, 46 40127 Bologna, Italy; maurizio.quartieri@unibo.it (M.Q.); enrico.muzzi@unibo.it (E.M.); moreno.toselli@unibo.it (M.T.)

[2] FArm MOnitoring Systems for Agriculture, Viale della Lirica, 11 48124 Ravenna, Italy; massimo.noferini@famosasrl.com

* Correspondence: elena.baldi7@unibo.it

**Abstract:** The aim of the present experiment was to evaluate the effectiveness of soil electrical conductivity (EC) measurement as a fast tool to assess mineral nitrogen (N) in orchards, in order to define precise N inputs that can help farmers to reduce useless fertilizer application. During one vegetative season, seven orchards of different species, supplied with mineral or organic fertilization, were monitored. Nitrate soil concentration was measured monthly by laboratory analyses, while soil EC and moisture were recorded continuously by soil probes. Nitrate and EC were positively correlated, laying the foundation for the identification of a fast and reliable index. However, while some dates showed a high Pearson correlation coefficient, no correlation was found for others. The correlation was not affected by type of fertilizer, and was higher in silty-clay-loam than in loam soils. Pooling all of the data, a significant correlation with a Pearson coefficient of 0.75 was found. The soil optimal nitrate N availability was defined by an EC in the range of 0.3 to 0.6 mS cm$^{-1}$. Although these are only preliminary results, our data are promising, showing a good suitability of soil EC measurement as a means to monitor soil mineral N availability.

**Keywords:** orchard nutrition; precision agriculture; soil texture; nitrate-N; soil probes; organic fertilizer

## 1. Introduction

The intensification of agricultural production systems has resulted in a dramatic increase of fertilizer inputs [1]. In the last 15 years (2002–2017), worldwide, use of nitrogen (N) fertilizers has increased from 83 Mt to 109 Mt, mainly due to the sharp rise of food production in China [2]. Although N is a fundamental nutrient for crop yields and food production, its inputs into agricultural ecosystems can cause water and air pollution as well as global warming and stratospheric ozone depletion. The EU launched several directives to reduce water pollution resulting from N from agricultural sources (EC-Council Directive, 1991) that led to the stabilization of N consumption around 11 Mt. In Italy, N use in agriculture decreased from 845 kt in 2002 to 602 kt in 2017 [1]. However, there is still broad room for improvement since farmers currently apply N according to general recommendation (i.e., Integrated Crop Management guidelines) without checking the real soil availability.

Precision agriculture, defined as "that kind of agriculture that increases the number of correct decisions per unit area of land per unit time with associated net benefits [3]", is among the most effective means of achieving sustainable agriculture goals. The introduction of precision agriculture techniques through wireless remote-control solutions offers great potential for improving fertilizer

use efficiency; however, it requires reliable tools capable of providing information on soil nutrient availability and plant nutritional status. Precision agriculture can assist farmers, since it permits the accurate and optimized use of inputs adapted to each single plant and soil status; consequently, it can lead to an increase of economic and environmental sustainability.

The evolution of measurement techniques that are capable of monitoring soil water contents in field and ionic solute distributions in the same sampling volume and in real time will improve our understanding of flow and transport processes of nutrients in the field and, consequently, will increase changes for effective fertilizer management. The use of probes that are able to monitor soil moisture and electrical conductivity (EC) at different depths, as well as nutrients in soil solution or in xylem sap, are some of several methods that can be used to reach these goals. However, the development of a decision support system able to interpret probe data and provide useful information to farmers will be needed.

Electric conductivity measures the presence of the major inorganic solutes dissolved in the aqueous phase consisting of soluble and readily dissolvable salts in soil solution, including cations (e.g., $Na^+$, $K^+$, $Mg^{2+}$, $Ca^{2+}$), anions ($Cl^-$, $HCO_3^-$, $NO_3^-$ and $SO_4^{2-}$), and nonionic solutes [4]. As a result, soil EC has become one of the most frequently used measurement characterizing field nutrient variability for application to precision agriculture [5]. A survey of six typical soils in the arid Southwest of the USA described how texture, water content, bulk density and organic matter interacted to influence the EC signal data and could build predictive maps of soil salinity [4]. In the area of investigation, characterized by lime or loam to silty-clay-loam soils, a high cation exchange capacity (CEC) reduced the number of ionic species in solution, with nitrate ($NO_3^-$) prevailing over the others. Nitrate concentration varied widely as a consequence of a complex interactions between microbiological, chemical and physical processes and fluctuated over time as a consequence of mineralization, immobilization and other turnover processes. The knowledge of nitrate concentration is important in developing management strategies for the efficient use of N by plants [6,7] and the reduction of $NO_3^-$-N leaching and consequent ground water contamination [8]. Since real-time continuous measurement of $NO_3^-$-N concentrations is required, the use of probes able to measure EC, and the correlation between EC and nitrate in the soil, could be a valuable solution to avoid excess N fertilizer supply.

Studies conducted in controlled conditions [9] or in field experiments [10–12] evidenced a direct relationship between EC and nitrate concentration in soil solutions. Moreover, the measurement of EC and $NO_3^-$ in different soil types [13], crops and farming systems showed a $R^2$ value of 0.98, supporting the hypothesis that the information provided by EC measurements is equivalent to that given by nitrate determinations. Based on the need to find a fast way to measure soil fertility, this study aimed to evaluate the effectiveness of soil EC as a fast and reliable tool to assess mineral N in orchard soils with different fertilization managements, in order to define precise N inputs and reduce fertilizer consumption.

## 2. Material and Methods

### 2.1. Orchard Description

The study was conducted in 2019 on 7 orchards that included 7 different species: apricot (cv. Faralia®), plum (cv. September Yummy), nectarine (cv. Romagna Red), pear (cv. Abbé Fetél), apple (cv. Rosy Glow), kiwifruit (cvs. Hayward and Dorí). The area of the experiment is characterized by a temperate climate; total precipitation in the period of the experiment (from April 26th to September 27th) was 305 mm for apricot and plum, 273 mm for peach, pear and apple, and 188 mm for both kiwifruit. The average temperature was 14.8 °C for apricot and plum, 21.3 °C for peach, pear and apple, and 21.6 °C for kiwifruit.

The main characteristics of each orchard are described in Table 1, and the main soil properties are reported in Table 2.

**Table 1.** Fruit species, variety, rootstock, year of plantation and planting density of the orchards.

| Species | Variety | Rootstock | Planting Year | Planting Distance (m) |
|---|---|---|---|---|
| *Prunus domestica* L. | September Yummy | GF677 (*P. persica* × *P. dulcis*) | 2015 | 4 × 1.5 |
| *Prunus armeniaca* L. | Faralia® | Mirobalan 29C (*P. cerasifera*) | 2014 | 4.8 × 2.5 |
| *Prunus persica* Batsch var. *nucipersica* | Romagna Red | Ishtara® (*P. persica* × *P. cerasifera*) | 2012 | 3.5 × 1.2 |
| *Malus domestica* Borkh | Rosy Glow | M9 | 2012 | 3.5 × 1 |
| *Pyrus comunis* L. | Abbé Fetél | Self-rooted | 2001 | 3.9 × 2.2 |
| *Actinidia chinensis chinensis* | Dorì | Hayward | 2017 | 4.5 × 2 |
| *Actinidia chinensis deliciosa* | Hayward | Self-rooted | 2016 | 5 × 2 |

**Table 2.** Soil texture, total N, organic matter (OM), cation exchange capacity (CEC) and world reference base (WRB) soil classification of the orchards at planting.

| Variety | Sand (%) | Loam (%) | Clay (%) | Total N (‰ DW[z]) | OM (% DW) | CEC (Meq 100 g$^{-1}$) | Texture | WRB |
|---|---|---|---|---|---|---|---|---|
| Apricot | 32 | 48 | 20 | 1.67 | 2.37 | 23.1 | loam | Fluvic Cambisols |
| Plum | 32 | 48 | 20 | 1.67 | 2.37 | 23.1 | loam | Fluvic Cambisols |
| Pear | 35 | 53 | 12 | 1.37 | 2.03 | 21.2 | silty-loam | Fluvic Cambisols |
| Peach | 16 | 57 | 27 | 1.12 | 1.43 | 14.9 | silty-clay-loam | Fluvic Cambisols |
| Apple | 42 | 43 | 15 | 1.27 | 1.82 | 14.5 | loam | Fluvic Cambisols |
| Dorì kiwifruit | 18 | 46 | 36 | 1.34 | 1.83 | 23.5 | silty-clay-loam | Hypovertic Cambisols |
| Hayward kiwifruit | 18 | 46 | 36 | 1.34 | 1.83 | 23.5 | silty-clay-loam | Hypovertic Cambisols |

[z]DW = soil dry weight. Analysis performed by an external laboratory according to official methods of chemical soil analysis (D.M. 13-09-1999).

Each orchard was divided into two big plots, of 2 to 6 rows of trees each and 50–100 trees per row, that were discriminated by the type of fertilization: one plot was supplied only with mineral fertilizers, the other amended with organic fertilizers (compost or cow manure) with rates defined in accordance to integrated Crop Management Guidelines of the Emilia-Romagna region [14]. In detail, organic amendment was applied in autumn, at the rate of 10 t FW ha$^{-1}$, and tilled at a depth of 0.25 m. Nitrogen application rate was 100 kg ha$^{-1}$ for apricot and peach, 90 kg ha$^{-1}$ for plum and pear, 80 kg ha$^{-1}$ for apple, 120 kg ha$^{-1}$ for green kiwifruit and 150 kg ha$^{-1}$ for yellow kiwifruit. Phosphorous was applied at 40 kg ha$^{-1}$ in peach, apricot and plum, 30 kg ha$^{-1}$ in apple and pear, 50 kg ha$^{-1}$ in green kiwifruit, and 60 kg ha$^{-1}$ in yellow kiwifruit. Potassium was supplied at 100 kg ha$^{-1}$ for apricot, plum, peach and pear, at 50 kg ha$^{-1}$ for apple, at 130 kg ha$^{-1}$ for green kiwifruit, and 145 kg ha$^{-1}$ for yellow kiwifruit. Each big plot was divided into three subplots (replications) for sampling; data were elaborated according to a completely randomized experimental design, with 3 replications.

The trees were regularly watered during the vegetative season with a drip irrigation system according to the evapotranspiration rate measured every day with meteorological probes installed in each orchard.

## 2.2. Probes Installation and Characteristics

In April 2019, a geo-referentiated node was positioned in the tree row in each orchard, 10–15 cm from drippers, and probes were set at a depth of 0.10 m. A homemade node (iFarming, Italy) was an independent device connected to one or more sensors and able to transmit data to a platform. The monitoring system detected, among other things: ground and air temperature, heat indexes, EC, soil water potential and percentage of moisture, rainfall, air humidity and leaf wetness. EC and soil moisture were measured with volumetric probes (Watermark, GMR, Scandicci, Italy) with three separated electrodes able to transmit data every 15 min. The EC probes measured from 0 to 10,000 $\mu S\ cm^{-1}$ with a resolution of 10 $\mu S\ cm^{-1}$; the soil moisture was measured from 1% to 100% with a resolution of 0.01%.

## 2.3. Soil Sampling and Analysis

On April 26th, May 21st, June 26th, August 24th and September 27th, soil cores were collected at a depth of 0–0.40 m to measure $NO_3^{-}$-N soil concentration. Nitrate-N [15] was extracted from 10 g of soil by a solution of 100 mL of 10 mM $CaCl_2$; samples were shaken at 100 rpm for 1 h and, after soil sedimentation, the limpid solution was filtered, collected, and stored at −20 °C until analysis using an auto analyser (Auto Analyzer AA3; Bran + Luebbe, Norderstadt, Germany).

## 2.4. Laboratory Determination

Electric conductivity of ammonium nitrate ($NH_4NO_3$) solution (Sigma-Aldrich, St. Louis, MO, USA) at different N concentrations was measured according to the official methods of chemical soil analysis (D.M. 13-09-1999), in the laboratory with a portable conductivity meter (XS cond 110, Eutech instruments, Thermo Fisher Scientific Inc., Waltham, MA, USA) to evaluate a theoretical correlation between N and EC. On the same soil used for nitrate-N determination (sampled in April and May), EC was also measured. Briefly, 5 g of dry soil was mixed with 25 mL of deionized water; samples were shaken at 100 rpm for 2 h and left to stand overnight. EC was measured on the limpid solution with a portable conductivity meter.

## 2.5. Statistical Analysis

The effect of fertilizer was evaluated using the software SAS (SAS Institute Inc., Cary, North Carolina, USA); when analysis of variance showed a statistically significant ($p \leq 0.05$) effect of treatment, the Student Newman–Keuls (SNK) test separated the means. The Pearson correlation coefficient was employed to evaluate the relationship between soil EC measured by the probes and analytically in the lab. Correlation analysis was also run to estimate the linear relationship between soil $NO_3^{-}$-N concentration measured in the laboratory and the EC obtained with probes in the field, characterized by the fertilization strategy and soil texture.

## 3. Results

### 3.1. Soil Characteristics

The investigated soils showed a texture ranging from loam (3), silty-loam (1), to silty-clay-loam (3) (Table 2). Total N ranged between 1.12% for peach to 1.67% for plum. Organic matter was related to total N and ranged from 1.43% to 2.37%. Cation exchange capacity ranged from 14.5 meq $cm^{-1}$ in the soil of apple to 23.5 meq 100 $g^{-1}$ for 'Dorì' and 'Hayward' kiwifruit (Table 2). Based on these data, the soil was divided into two sets according to their texture: loam and silty-clay-loam for the correlation analysis.

### 3.2. Trends of Soil Nitrate-N and EC

On average, the $NO_3^--N$ soil concentration was not affected by the fertilization management technique and ranged between <10 mg N kg$^{-1}$ in May to <30 mg N kg$^{-1}$ in September (Table 3).

**Table 3.** Soil nitrate ($NO_3^--N$) concentration, electric conductivity (EC) and moisture during the experiment (values are average of data from all orchards).

| Sampling Date | Fertilization | NO$_3^-$-N (mg kg$^{-1}$ ss) | EC (mS cm$^{-1}$) | Moisture (%) |
|---|---|---|---|---|
| April 26th | Organic | 18.1 | 0.618 | 36.3 |
| | Mineral | 17.4 | 0.442 | 30.3 |
| *Significance* | | *ns$^z$* | *ns* | *ns* |
| May 21st | Organic | 7.77 | 0.466 | 36.3 |
| | Mineral | 6.16 | 0.389 | 31.9 |
| *Significance* | | *ns* | *ns* | *ns* |
| June 26th | Organic | 27.8 | 0.474 | 32.4 |
| | Mineral | 15.9 | 0.496 | 33.0 |
| *Significance* | | *ns* | *ns* | *ns* |
| August 24th | Organic | 10.6 | 0.375 | 27.6 |
| | Mineral | 9.88 | 0.400 | 31.3 |
| *Significance* | | *ns* | *ns* | *ns* |
| September 27th | Organic | 34.9 | 0.347 | 26.5 |
| | Mineral | 27.0 | 0.319 | 24.8 |
| *Significance* | | *ns* | *ns* | *ns* |

$^z$ns = effect of treatment not significant at $p \leq 0.05$.

For apricot, soil nitrate-N availability decreased from April until August for both mineral and organic fertilized soil, then it increased in September only in organic plots (Figure 1A).

In plum, soil $NO_3^--N$ ranged between 5 and 30 mg kg$^{-1}$ over the season for both fertilization management techniques (Figure 1C). In pear, $NO_3^--N$ increased in June (only in organically fertilized soil), August and September, reaching values higher than 60 mg kg$^{-1}$ (Figure 1E). In peach, soil $NO_3^--N$ was nearly steady, lower than 12 mg kg$^{-1}$ (Figure 1B). In apple, it was lower than 15 mg kg$^{-1}$ until September when it peaked at 40 mg kg$^{-1}$ in both fertilization treatments (Figure 1D). In yellow kiwifruit, soil nitrate-N availability showed three peaks higher than 50 mg kg$^{-1}$ in April, June and September (Figure 1F). Finally, in green kiwifruit, $NO_3^--N$ showed two peaks in June (when organic fertilizer induced a higher $NO_3^--N$ soil concentration than the mineral fertilizer) and September (Figure 1G).

In apricot, the EC ranged from 0.32 mS cm$^{-1}$ and 0.46 mS cm$^{-1}$ in mineral fertilized plots and between 0.32 mS cm$^{-1}$ and 0.6 mS cm$^{-1}$ in organically fertilized soil, while soil moisture ranged between 24.8% to 36.3% (Figure 2A). In the plum orchard, the EC values were higher at the beginning of the season (0.770 mS cm$^{-1}$) than in September (0.156 mS cm$^{-1}$ for mineral and 0.256 mS cm$^{-1}$ for organic); soil moisture ranged between 20% and 40% (Figure 2C). In peach, for both fertilizations, the EC was almost stable until the beginning of June (around 0.3 mS cm$^{-1}$), then it decreased and stabilized around 0.2 mS cm$^{-1}$; a similar trend was also observed for soil moisture, which was 30% at the beginning of the season and between 10% and 20% at the end of it (Figure 2B). In both fertilization plots of apple, the EC ranged between 0.3 and 0.4 mS cm$^{-1}$, while soil moisture was between 20% and 33% in mineral and 10–34% in organically fertilized soil (Figure 2D). In pear, EC ranged between 0.35 to 0.55 mS cm$^{-1}$ in mineral fertilized plots and between 0.30 to 0.45 mS cm$^{-1}$ in organically fertilized plots, while soil moisture ranged between 30% and 45% (Figure 2E). In mineral plots of yellow kiwifruit, EC was almost stable at the beginning of the season until mid-June, with values around 5 mS cm$^{-1}$, then it sharply decreased and stabilized around 0.6 mS cm$^{-1}$. On the other hand, EC in organic plots peaked twice in April and June (2.2 mS cm$^{-1}$), and from June 19th it remained almost steady (0.35 mS cm$^{-1}$). Soil moisture ranged between 20% to 50% (Figure 2F). In green kiwifruit, EC ranged between 0.3 and 0.4 mS cm$^{-1}$ both in mineral and organically fertilized plots; the soil moisture trend was similar in the two fertilization strategies, ranging from 20% to 40% (Figure 2G).

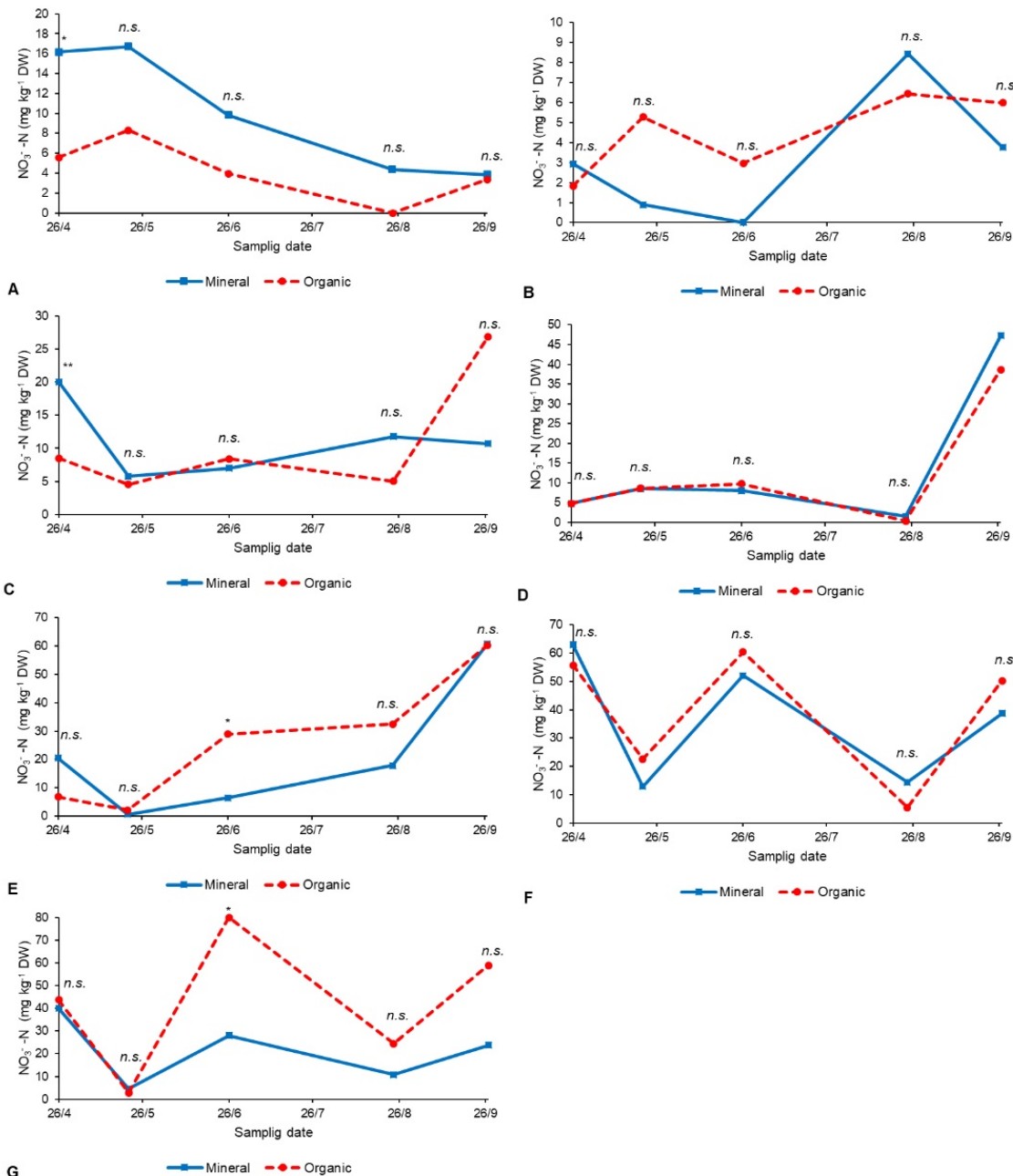

**Figure 1.** Effect of fertilization treatment on soil nitrate ($NO_3^-$-N) during the season in apricot (**A**), peach (**B**), plum (**C**), apple (**D**), pear (**E**), yellow kiwifruit (**F**) and green kiwifruit (**G**) orchard.

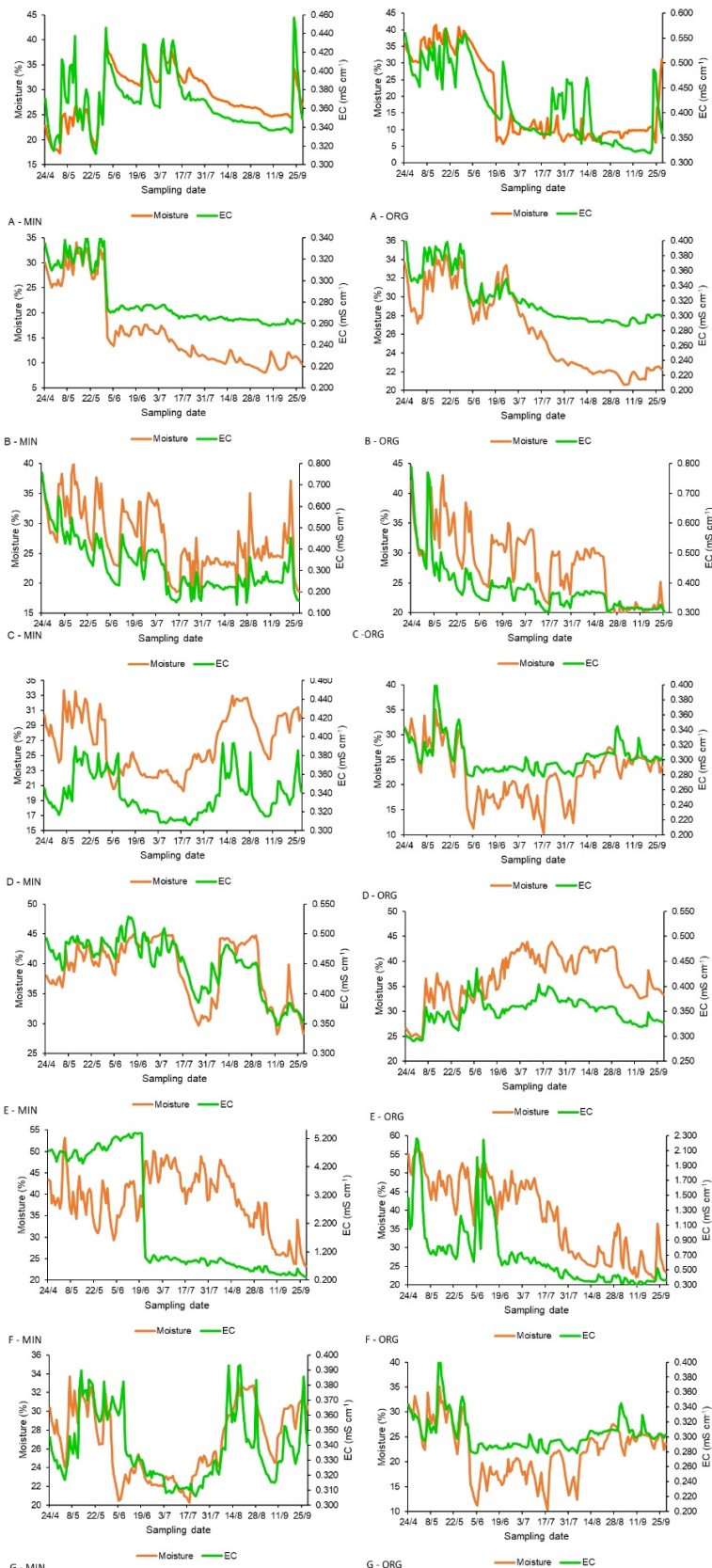

**Figure 2.** Trend of soil electric conductivity (EC) and moisture in mineral (MIN) and organic (ORG) fertilized plots in apricot (**A**), peach (**B**), plum (**C**), apple (**D**), pear (**E**), yellow kiwifruit (**F**) and green kiwifruit (**G**) orchards during the season.

### 3.3. Correlations

Laboratory analyses evidenced a positive correlation between N in water solution and EC (Figure 3). Moreover, the values of EC measured in the field with probes were linearly related (r = 0.67; $p < 0.1\%$) to values in solution extracted from soil sampled in April and May (data not shown).

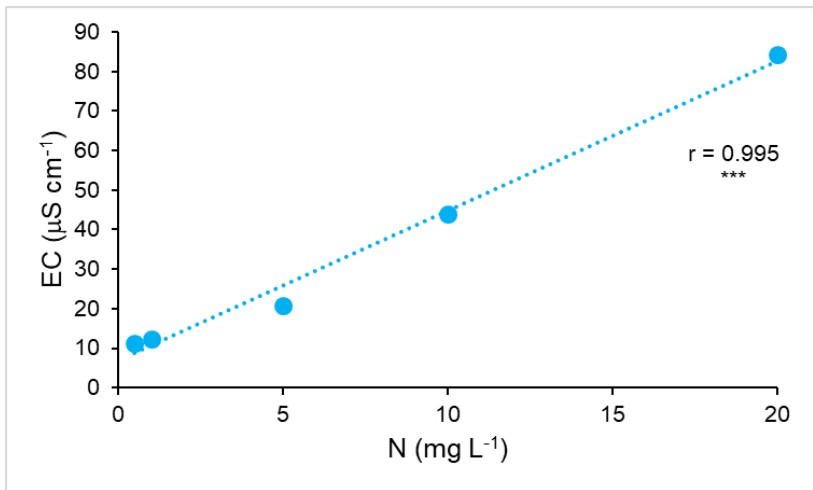

**Figure 3.** Correlation between N as $NH_4NO_3$ in water solution and electric conductivity (EC) measured in the laboratory. *** = effect of treatment significant at $p \leq 0.001$.

The soil EC and $NO_3^-$-N showed a significant correlation in April only for compost treated plots (Figure 4A) and in June for both treatments (Figure 4C). However, in May (Figure 4B) and August (Figure 4D), no significant correlations between EC and soil $NO_3^-$-N were observed.

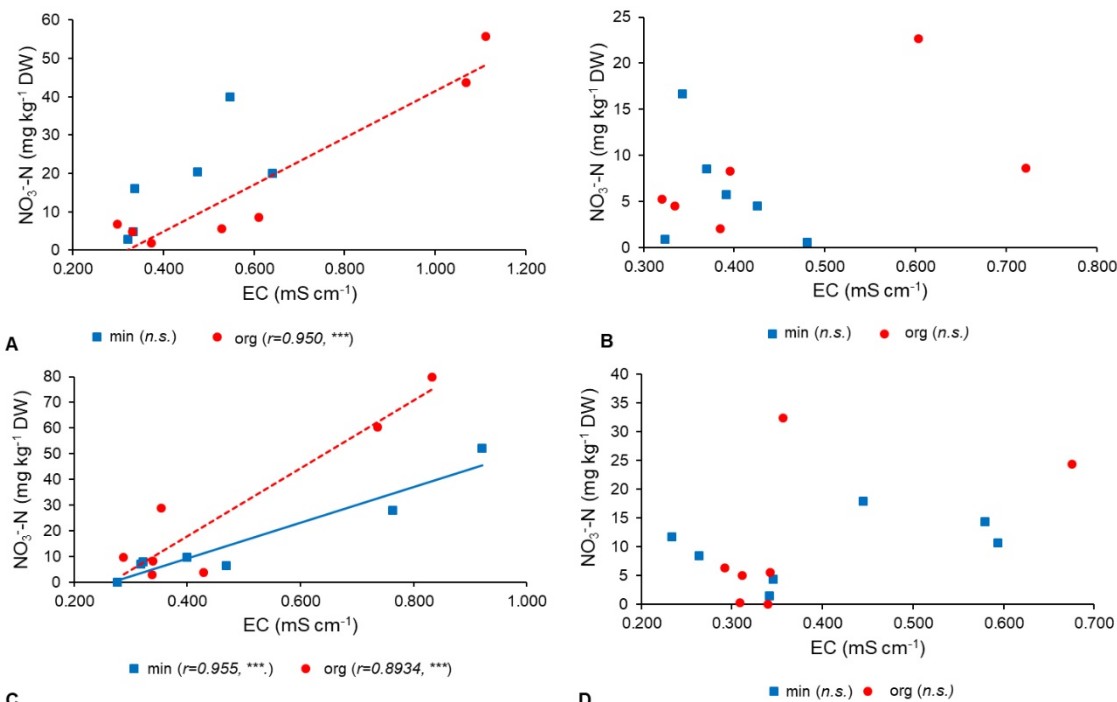

**Figure 4.** Correlation between electric conductivity (EC) measured by probes and soil nitrate ($NO_3^-$)-N concentration in organic and mineral fertilized plots in April (**A**), May (**B**), June (**C**) and August (**D**). ***, n.s. = effect of treatment significant at $p \leq 0.001$ or not significant, respectively.

The correlation analysis was similar for the two fertilization strategies, with the Pearson coefficients r = 0.758 for mineral and r = 0.749 for organic. Considering the two soil textures, unlike loam which was not statistically significant, in silty-clay-loam soils had a significant correlation between EC and $NO_3^--N$, with a Pearson coefficient of 0.81 (Figure 5).

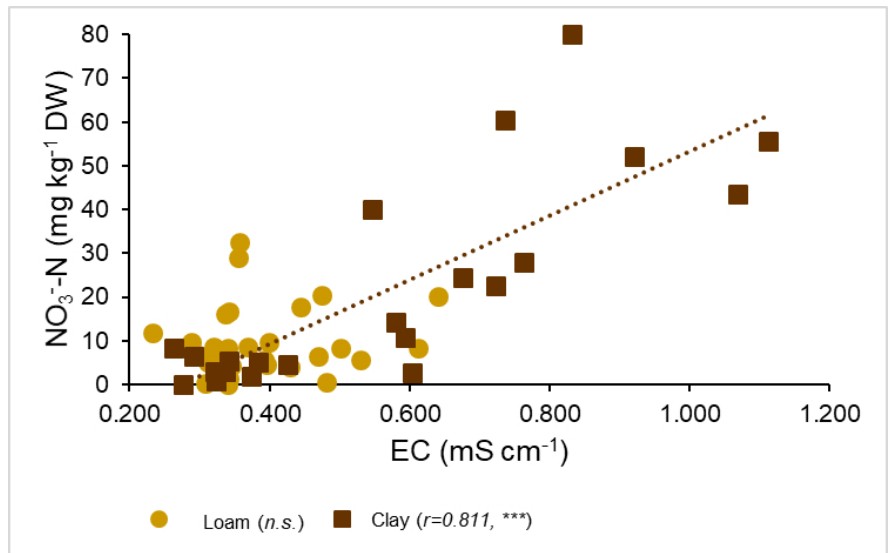

**Figure 5.** Correlation between electric conductivity (EC) measured by probes and soil nitrate ($NO_3^--N$) concentration in loam (circle) and silty-clay-loam soils (data from April to August). *** = effect of treatment significant at $p \leq 0.001$.

When all the sampling times and treatments were pooled together, soil EC measured in the field with the probes was correlated to soil $NO_3^--N$ as follows: $NO_3^--N$ (mg kg$^{-1}$) = EC (mS cm$^{-1}$) × 64.3 − 15.2 (Figure 6).

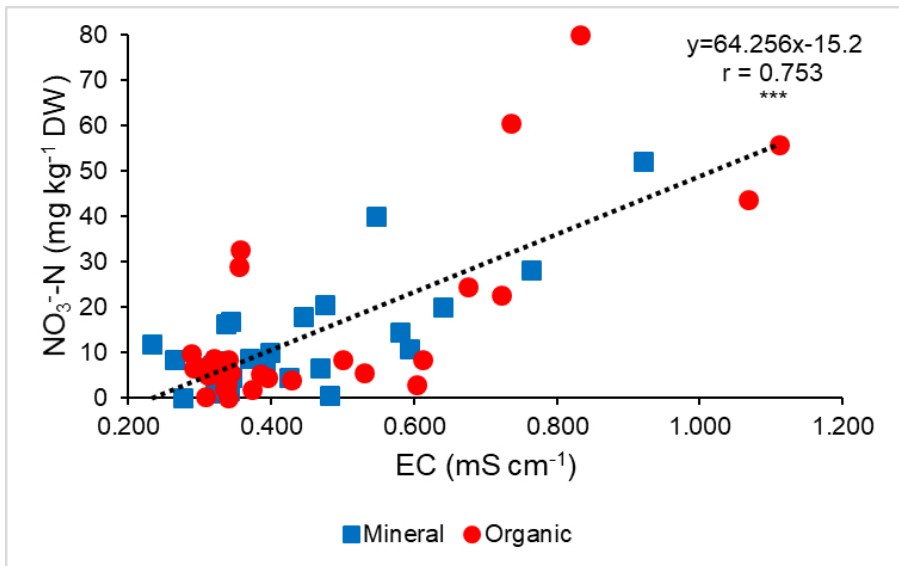

**Figure 6.** Correlation between electric conductivity (EC) measured by probes and soil nitrate ($NO_3^--N$) concentration in organic and mineral plots (data from April to August). *** = effect of treatment significant at $p \leq 0.001$.

## 4. Discussion

Orchard fertilization is the major tool affecting soil nutrient availability and, consequently, influencing yield and fruit quality. As a consequence, it is important to identify and develop useful strategies that are able to give reliable information on orchard nutritional status in real time.

Since N is the main element of plant fertilization, the measurement of soil mineral nitrogen concentration gives steady information on N availability. In the soil of the Po Valley, in northern Italy, mineral N is mainly comprised of $NO_3^-$-N, since $NH_4^+$-N oxidizes rapidly [16]. As a result, the mineral-N:$NO_3^-$-N ratio is almost steady, and only water soluble $NO_3^-$-N is commonly detected for fertilization management [17]. During the growing season, the optimal $NO_3^-$-N concentration ranges between 5–20 mg $kg^{-1}$ [18,19].

The improper use of N fertilizers, besides contamination of ground water, can have negative impacts on plants and lead to excess vegetative growth [20], increase susceptibility to diseases and physiological disorders [21], and decrease the quality and shelf-life of fruits [22].

Data obtained in a field experiment on peppermint [23] demonstrated that estimation of EC and nitrate concentrations using time domain reflectometry exhibited similar patterns, magnitudes and variance to those based on direct soil measurements, indicating that the employment of EC can be used to estimate ionic solute concentrations in agricultural fields. In addition, it was demonstrated [13] that the information gained from EC measurements was equivalent to that obtained from nitrate determinations, and consequently, the EC could be used as a simpler and less costly index of nutrient loss.

Our data showed that the values of EC provided good information on soil N availability using the equation that resulted from the correlation between EC and nitrate-N soil concentration. The same results were obtained in a lysimeter experiment [24] that showed a robust correlation (r = 0.780) between nitrate and EC, derived from analyzing 500 soil samples. In that study, the relationship was best described by a second degree ($y = 84.801x^2 - 10.059x$) regression equation, while we found a linear correlation. The difference observed in our study was probably related to the lower fertility of our soils that showed a lower $NO_3^-$-N than that investigated by Koumanov and co-workers [24].

We want to stress that the relationship observed in our study was the result of the pooling of the data collected during the whole season. However, when the correlation was considered for each specific phenological stage, a relationship between EC and $NO_3^-$-N was not always found. Further research to investigate the interference of factors such as soil texture, soil CEC, salinity of water, pH, etc., are needed. In fact, we observed that the correlation was stronger in clay than in loam soils, likely because of the higher retention strength that removed the potential interference of most of the cations in solution. On the other hand, the type of fertilization (mineral vs. organic) did not modify the relationship between $NO_3^-$-N and EC. This may have been because the fertility of the investigated soils was high and the addition of mineral N probably showed a short-term effect on mineral N availability [25] that was not detected by EC probes. In addition, a high soil CEC probably limited the effect of organic fertilizer on soil ion retention. Under our experimental conditions, no water-soluble cations were found in soil water extracts (data not reported); however, in soil with a low pH and sandy texture, it is possible to find the interference of other ions.

To avoid the effect of soil moisture on $NO_3^-$-N dilution, we analyzed the soils at their field capacity; however, water from irrigation or rain has the potential to modify the soluble solids concentration in soil solutions by dilution or concentration according to the quality of the water itself.

## 5. Conclusions

Our aim to use the EC to estimate nitrate-N in the soil and provide a fast tool for N evaluation during the growing season was partially satisfied. Electric conductivity seemed to estimate soil $NO_3^-$-N in soils with high clay content, no matter the fertilization strategy. However, these results are still preliminary and more research should be conducted in order to better standardize the relationship so that farmers can adjust fertilizer application according to the crop needs.

**Author Contributions:** Conceptualization, M.T. and E.B.; Field sampling, M.T. and M.Q.; Laboratory analysis, E.B. and M.Q.; Statistical analysis, E.M.; Software and probes, M.N.; writing—original draft preparation, M.T. and E.B. All authors have read and agreed to the published version of the manuscript.

**Funding:** The project was financed by the Emilia-Romagna Region; P.S.R. 2014-2020-16.2.01; "Toward digitalization of fruit production (DIGIFRUIT)".

**Acknowledgments:** The authors would like to thank Andrea Ghirotti and Lorenzo Donati for their assistance in collecting samples, and the farms Baccarini Barbara, Donati Giuliano and Zani Maurizio for hosting the experiment.

**Conflicts of Interest:** The authors declare no conflict of interest.

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
