# Peer review of "Use of In Situ Soil Solution Electric Conductivity to Evaluate Mineral N in Commercial Orchards: Preliminary Results"

_horticulturae, doi:10.3390/horticulturae6030039_

Round 1
Reviewer 1 Report
- It is a pity that the authors limited their studies to one growing season.
- The nitrogen symbol is incorrectly spelled in the title of the manuscript.
- Since the manuscript needs improvement, the abstract will require substantial changes.
- The Introduction needs to be strengthened to outline previous research and to build a better justification for this manuscript.
- Please add to Material and methods section:
- data regarding EC and N-NO3 to the table 2.
-information about soil types in the orchards, weather /irrigation, the size of each plot, the rates of mineral and organic fertilization, sampling dates, references to the all methods used
- Please move information about experimental design from 2.5. section to 2.1. section.
- Results should be presented separately for each orchard in the table (table 3) or figures 1, 2.
- The conclusions do not fully relate to the aim of the paper
- Please provide more references in the Introduction and Discussion sections.
- The paper must be revised by a native English speaker.
Reviewer 2 Report
Broad Comments:
- The “Introduction” section doesn’t describe the problem in detail and the need for research. Also lacks strong literature review. There are too many general conceptual statements that need references.
- The field experimental design is not clear. The amount of fertilizer and manure applied should be clearly stated. Please see comments section
- Discussion section: It is understood that this is preliminary results. But there are some broad concepts and correlations discussed in this section which requires evidence from data. For example, plant available N is not measured and plant nutritional demand for different species not included in the experimental design. However, the authors discussed these correlations in the manuscript extensively.
Specific Comments
Introduction
Line 27: Consider revising “is facing”
Line 28: This implies “that” the intensification
Line 29: Need reference for this statement.
Line 42: reliable tools capable “of providing” information on soil nutrient availability. Consider changing wherever you used “nutrient soil” to “soil nutrients”
Line 45: Consider field soil moisture contents
Line 52: Consider revising this sentence. Electric conductivity “measures the presence”
Line 55: Consider “field nutrient variability”
Line 58-60: In lime?? Consider revising this whole sentence. It is confusing.
Line 61: Consider removing “useless” from the fertilizer consumption. Reducing fertilizer consumption implies the same and this is not the proper usage.
Materials and Methods:
Table 2: Standard usage “Cation exchange capacity (CEC)”
Line 75: On what basis the manures were applied? Were they matched to mineral fertilizer requirements based on soil tests?
Results:
Figure 1: “Sampling data” to “Sampling date”. Make this change thought the manuscript.
Figure 4: This correlation is not very strong and trendline is skewed to few data points.
Discussion:
Line 189: You only measured soil extractable nutrients. They cannot be interpreted as “plant available nutrients”.
Line 190-191: Each tree in the orchard has a different pattern of N requirement at different growth stages. Did you /plan to measure those?
Line 199: rate of N? Uptake or rate of N applied?
Line 200: How did you come up with an optimal N availability? How and for what species is this range that you recommend ?
Conclusion
Line 223: “root uptake kinetics”. Is this measured in this experiment or planned for future? Please be clear what has been done and what you planned for future.
Reviewer 3 Report
The experimental data presented by the authors are insufficient for an unambiguous statement that the increase in soil solution electric conductivity caused by an increase in the content of mineral nitrogen.
- The Introduction (lines 53-54) states that electrical conductivity refers to the presence of the major inorganic solutes, including cations and anions. In this regard, it is unclear why the authors did not study these substances, although in lines 208-210 of the Discussion written that these studies will be performed in the future.
- Materials and methods (line 75): more detailed data should be given on what fertilizers were applied and in what doses.
- Speaking about organic fertilization (compost or cow manure), the authors do not focus on the fact that in addition to nitrogen, these types of fertilizers are characterized by a high content of potassium (as a rule, exceeds the content of nitrogen), calcium and phosphorus. Therefore, it is incorrect to associate an increase in electric conductivity only with nitrogen.
- The manuscript lacks a clear explanation for analysis of nitrogen content in nitrate form only. The explanation in lines 58-60 is not convincing, since it is well known that the nitrogen stock in cow manure is not limited to nitrate and ammonium forms.
- The table 3 indicates the absence of significant differences in nitrogen content and electric conductivity between organic and mineral fertilization. This is somewhat strange. As a rule, the nitrogen fertilization gives a noticeable increase in the nitrogen content in the soil. Without data on fertilizer doses, this is impossible to understand.
- For correlation search between electric conductivity and nitrogen content, authors used a solution of NH4NO3. Moreover, the data obtained are extrapolated only to nitrogen in nitrate form, and the ammonium form of nitrogen is not taken into account at all.
- To explain the data on changes in the nitrogen content in the soil under various fruit species, it would be logical to characterize nutritional requirements of these plant species.
- The meaning of Figure 4 is not clear.
- The Discussion section contains virtually no references.
- Minor inaccuracies:
- abstract: the term “electric conductivity” should be given before “EC” abbreviation;
- please, use the “cation exchange capacity (CEC)” term instead of “exchange cation capacity (ECC)”.
- line 95: “Germany” instead of “Germania”.
Round 2
Reviewer 1 Report
1. The Authors have not followed all the comments and suggestions I made in the first review.
2. The data regarding EC and N-NO3 is missed (table 2).
3. Language and editing errors still occur in the revised version, e.g. sampling data (Figure 1) or ‰ (table 2). Please write down the nitrate-nitrogen as NO3-N or NO3--N.
4. It would be advisable for the Authors to present information in the manner typical of scientific papers. i.e. information about experimental design should be put next to the treatment description.
5. The Authors did not provide information on soil types according to World Reference Base for Soil Resources.
6. The results should be uniformly presented and not intentionally selected by the authors.
7. The period (only one year) and a scope of the conducted experiment do not allow drawing too far-reaching statements in the discussion and conclusion sections.
Author Response
Dear reviewer,
thanks for your comments.
Below our answers to the points you raised.
- The data regarding EC and NO3-N is missed (table 2).
REPLY: Table 2 shows stable chemical and physical properties of investigated soils, that do not change in the short period; on the other hand, as shown in figure 1 and 2, EC and NO3-- N change during the vegetative season and cannot be indicated as a general feature of the soil. Most of all, our manuscript is based on the seasonal variation of EC and NO3, so that it is not possible to indicate 1 single value. Since data of EC and NO3—N are deeply presented in table 3, figure 1 and 2, we consider a wrong repetition to show them also in table 2
- Language and editing errors still occur in the revised version, e.g. sampling data (Figure 1) or ‰ (table 2). Please write down the nitrate-nitrogen as NO3-N or NO3--N.
REPLY: we corrected mistakes.
- It would be advisable for the Authors to present information in the manner typical of scientific papers. i.e. information about experimental design should be put next to the treatment description.
REPLY: done.
- The Authors did not provide information on soil types according to World Reference Base for Soil Resources.
REPLY: done.
- The results should be uniformly presented and not intentionally selected by the authors.
REPLY: we modified figure 2 and presented EC and soil moisture values for all the orchards.
- The period (only one year) and a scope of the conducted experiment do not allow drawing too far-reaching statements in the discussion and conclusion sections.
REPLY: we corrected it.
Reviewer 2 Report
I see that the authors have incorporated most of the suggested revisions. I would recommend to accept the article in the present form.
Author Response
Thank you
Reviewer 3 Report
Dear authors,
I am satisfied with your answers.
Author Response
Thank you
Round 3
Reviewer 1 Report
The authors have addressed the comments and suggestions I made in the review. I have one more suggestion:
When I wrote in the last review: It would be advisable for the Authors to present information in the manner typical of scientific papers. i.e. information about experimental design should be put next to the treatment description, I meant one fragment of the text:
according to a complete randomized experimental design, with 3 replications.
Please leave it in the section 2.1 Orchard description. The rest of the paragraph (L110-116) should be placed where it was in the previous version of the manuscript, i.e. in the section – 2.5. Statistical analysis
Author Response
Dear reviewer,
thanks again for your comments. We corrected section 2.1 as you suggested.
Best regards
Elena Baldi